



# Multiscale observations of NH$_3$ around Toronto, Canada

Shoma Yamanouchi[1], Camille Viatte[2], Kimberly Strong[1], Erik Lutsch[1], Dylan B. A. Jones[1],
Cathy Clerbaux[2], Martin Van Damme[3], Lieven Clarisse[3], and Pierre-Francois Coheur[3]

[1]Department of Physics, University of Toronto, Toronto, Ontario, Canada
[2]LATMOS/IPSL, Sorbonne Université, UVSQ, CNRS, Paris, France
[3]Université Libre de Bruxelles (ULB), Spectroscopy, Quantum Chemistry and Atmospheric Remote Sensing (SQUARES),
Brussels, Belgium

**Correspondence:** Shoma Yamanouchi (syamanou@physics.utoronto.ca)

**Abstract.** Ammonia (NH$_3$) is a major source of nitrates in the atmosphere, and a major source of fine particulate matter. As such, there have been increasing efforts to measure the atmospheric abundance of NH$_3$ and its spatial and temporal variability. In this study, long-term measurements of NH$_3$ derived from multiscale datasets are examined. These NH$_3$ datasets include 16 years of total column measurements using Fourier transform infrared (FTIR) spectroscopy, three years of surface in-situ mea-

surements, and 10 years of total column measurements from the Infrared Atmospheric Sounding Interferometer (IASI). The datasets were used to quantify NH$_3$ temporal variability over Toronto, Canada. The multiscale datasets were also compared to assess the observational footprint of the FTIR measurements.

All three time series showed positive trends in NH$_3$ over Toronto: $3.34 \pm 0.46$ %/year from 2002 to 2018 in the FTIR columns, $8.88 \pm 2.83$ %/year from 2013 to 2017 in the surface in-situ data, and $8.38 \pm 0.77$ %/year from 2008 to 2018 in the

IASI columns. To assess the observational footprint of the FTIR NH$_3$ columns, correlations between the datasets were examined. The best correlation between FTIR and IASI was obtained with coincidence criteria of $\leq 25$ km and $\leq 20$ minutes, with $r = 0.73$ and a slope of $1.14 \pm 0.06$. Additionally, FTIR column and in-situ measurements were standardized and correlated. Comparison of 24-day averages and monthly averages resulted in correlation coefficients of $r = 0.72$ and $r = 0.75$, respectively, although correlation without resampling to reduce high-frequency variability led to a poorer correlation, with $r = 0.39$.

The GEOS-Chem model, run at $2° \times 2.5°$ resolution, was compared against FTIR and IASI to assess model performance and investigate correlation of observational data and model output, both with local column measurements (FTIR) and measurements on a regional scale (IASI). Comparisons on a regional scale (a domain spanning 35°N to 53°N, and 93.75°W to 63.75°W) resulted in $r = 0.57$, and thus a coefficient of determination, which is indicative of the predictive capacity of the model, of $r^2 = 0.33$, but comparing a single model grid point against the FTIR resulted in a poorer correlation, with $r^2 = 0.13$,

indicating that a finer spatial resolution is needed for modeling NH$_3$.



# 1 Introduction

Ammonia ($NH_3$) in the atmosphere plays an important role in the formation of nitrates and ammonium salts, is a major pollutant, and is known to be involved in numerous biochemical exchanges affecting all ecosystems (Erisman et al., 2008; Hu et al., 2014). As one of the main sources of reactive nitrogen in the atmosphere, $NH_3$ is also associated with acidification and eutrophication of soils and surface waters, which can negatively affect biodiversity (Vitousek et al., 1997; Krupa, 2003; Bobbink et al., 2010). Furthermore, $NH_3$ reacts with nitric acid and sulfuric acid to form ammonium salts, which are known to account for a large fraction of particulate matter (Schaap et al., 2004; Pozzer et al., 2017). Particulate matter, especially that smaller than 2.5 microns ($PM_{2.5}$), poses serious health hazards and is a major contributor to smog, affecting life expectancy in the United States (Pope et al., 2009) and globally (Giannadaki et al., 2014).

Due to the negative impacts $NH_3$ can have on public health and the environment, $NH_3$ emissions are regulated in some parts of the world (e.g., the 1999 Gothenburg Protocol to Abate Acidification, Eutrophication and Ground-level Ozone). However, global $NH_3$ emissions are increasing (Warner et al., 2016; Lachatre et al., 2019), and this has been attributed to increases in agricultural livestock numbers and increased nitrogen fertilizer usage (Warner et al., 2016). In addition, as the world population continues to grow, and the demand for food rises, $NH_3$ emissions are expected to further increase (van Vuuren et al., 2011). Observations show increasing abundance of atmospheric $NH_3$, particularly in the Eastern United States, where trends as high as 12 % annually were observed (Yu et al., 2018). Yu et al. (2018) concluded that the increase in $NH_3$ abundance was in part due to decreasing $SO_2$ and nitrous oxides, owing to more stringent emissions regulations.

Atmospheric $NH_3$ is rapidly removed by wet and dry deposition as well as by reactions with acids in the atmosphere, and thus has a relatively short lifetime ranging from a few hours to a few days (Galloway et al., 2003; Dammers et al., 2019). $NH_3$ lifetime may be longer for certain cases, such as biomass burning emissions that inject $NH_3$ into the free troposphere, attenuating depositional and chemical losses, although physical and chemical mechanisms that lead to transport of $NH_3$ in biomass burning plumes over long distances remain uncertain (Lutsch et al., 2016, 2019b). Dependence of $PM_{2.5}$ formation on $NH_3$ over urban areas also remains uncertain, and atmospheric chemical transport models have difficulty simulating $NH_3$ and $PM_{2.5}$ (e.g., Van Damme et al., 2014; Fortems-Cheiney et al., 2016; Schiferl et al., 2016; Viatte et al., 2020).

Modeled $NH_3$ can be used to supplement observations (Liu et al., 2017). However, ground-level $NH_3$ abundances are poorly modeled, due to coarse model resolution, uncertain emissions inventories, and simplification of chemistry schemes (Liu et al., 2017). Additionally, long-term trend analyses using models have been sparse (Yu et al., 2018). Representative measurements of $NH_3$ on both local and regional scales, as well as their spatiotemporal variabilities, are needed to better understand and model $NH_3$ and $PM_{2.5}$ formation (Viatte et al., 2020).

Toronto is the most populous city in Canada, and $NH_3$, along with other pollutants, is monitored by several instruments. In particular, the Fourier transform infrared (FTIR) spectrometer situated at the University of Toronto Atmospheric Observatory (TAO) has been making regular measurements since 2002. It is located in downtown Toronto, where local point sources (i.e. vehicle emissions), as well as nearby agricultural emissions, are major sources of $NH_3$ (Zbieranowski and Aherne, 2012; Hu et al., 2014; Wentworth et al., 2014). Toronto is also regularly affected by biomass burning plumes transported from the USA





and other regions of Canada (Griffin et al., 2013; Whaley et al., 2015; Lutsch et al., 2016, 2019a). As such, the time series of total column $NH_3$ measured at TAO exhibits long-term trends and pollution episodes. Toronto also has in-situ (surface) measurements of $NH_3$ (Hu et al., 2014), made by Environment and Climate Change Canada. A study by Hu et al. (2014) investigating $NH_3$ in downtown Toronto has shown that greenery within the city is an important source of $NH_3$ when temperatures

are above freezing, and that potential sources at temperatures below freezing have yet to be investigated. Additionally, recent studies have shown the increased capacity for satellite-based instruments to measure spatial and temporal distributions of $NH_3$ total columns at global (Van Damme et al., 2014; Warner et al., 2016; Shephard et al., 2020), regional (Van Damme et al., 2014; Warner et al., 2017; Viatte et al., 2020), and point-source scales (Van Damme et al., 2018; Clarisse et al., 2019a; Dammers et al., 2019).

In this study, $NH_3$ variability over Toronto is investigated using ground-based FTIR data, in-situ measurements, and satellite-based observations from the Infrared Atmospheric Sounding Interferometer (IASI). This study is a part of the AmmonAQ project, which investigates the role of $NH_3$ in air quality in urban areas. Trends in the $NH_3$ time series and their statistical significance are determined, and the GEOS-Chem model is also used to supplement and compare against observations. Additionally, correlations between FTIR and IASI, FTIR and in-situ, in-situ and IASI, FTIR and model data, as well as IASI and

model data on a regional scale are analyzed to assess the observational footprint of the FTIR $NH_3$ measurements.

The paper is organized as follows: Section 2 describes the FTIR retrieval methodology, the in-situ and satellite data, the GEOS-Chem model, and the analysis methodologies. Section 3 presents the trend analysis of the FTIR, in-situ, and IASI measurements, the results of the correlation studies, and the FTIR $NH_3$ observational footprint analysis. Section 4 presents the evaluation of the GEOS-Chem model, and conclusions are provided in Section 5.

## 2   Datasets and Methods

### 2.1   FTIR Measurements

Ground-based $NH_3$ total columns used in this study were retrieved from infrared solar absorption spectra recorded using an ABB Bomem DA8 FTIR spectrometer situated at the University of Toronto Atmospheric Observatory in downtown Toronto, Ontario, Canada (43.66°N, 79.40°W, 174 masl). This instrument has been making measurements since mid-2002, and trace

gas measurements are contributed to the Network for Detection of Atmospheric Composition Change (NDACC; http://www.ndsc.ncep.noaa.gov/) (De Mazière et al., 2018). The DA8 has a maximum optical path difference of 250 cm, with a maximum resolution of 0.004 cm$^{-1}$, and is equipped with a KBr (700-4300 cm$^{-1}$) beamsplitter. While the FTIR is equipped with both InSb and HgCdTe (MCT) detectors, $NH_3$ profiles were retrieved using the MCT detector, which is responsive from 500-5000 cm$^{-1}$. The DA8 is coupled to an active sun-tracker, which was manufactured by Aim Controls. The tracker is driven by two

Shinano stepper motors on elevation and azimuth axes. The active tracking was provided by four photo-diodes from 2002-2014. This was upgraded to a camera and solar-disk-fitting system in 2014. Detailed specifications of the system can be found in Wiacek et al. (2007). Due to the nature of solar-pointing FTIR spectroscopy, the measurements are limited to sunny days, resulting in gaps in the time series. Measurements are typically made on 100-150 days per year.



The TAO FTIR uses six filters recommended by the NDACC Infrared Working Group (IRWG), and measures spectra through each filter in sequence. $NH_3$ profiles were retrieved using two microwindows of 930.32-931.32 $cm^{-1}$ and 966.97-967.675 $cm^{-1}$. Interfering species include $H_2O$, $O_3$, $CO_2$, $N_2O$ and $HNO_3$. The solar absorption spectra recorded by the DA8 were processed using the SFIT4 retrieval algorithm (https://wiki.ucar.edu/display/sfit4/). SFIT4 uses the optimal estimation method (OEM) (Rodgers, 2000), and works by iteratively adjusting the target species volume mixing ratio (VMR) profile until the difference between the calculated spectrum and the measured spectrum, and the difference between the retrieved state vector and the a priori profile is minimized. The calculated spectra use spectroscopic parameters from HITRAN 2008 (Rothman et al., 2009), and atmospheric information (temperature and pressure profiles for any particular day) provided by the US National Centers for Environmental Prediction (NCEP). A priori VMR profiles were obtained from balloon-based measurements (Toon et al., 1999). The $NH_3$ retrieval methodology used at TAO is described in detail in Lutsch et al. (2016).

Uncertainties in the retrievals include measurement noise and forward model errors. Smoothing errors that arise due to the discretized vertical resolution were not included, to conform to NDACC standard practice. Measurement noise error includes errors due to uncertainties in instrument line shape, interfering species, and wavelength shifts. Uncertainties in line intensity and line widths were calculated based on HITRAN 2008 errors. Error analysis was performed on all retrievals (following Rodgers, 2000); the resulting errors were grouped into random and systematic uncertainties, and added in quadrature. The resulting mean uncertainties averaged over the entire time series, were 12.9% and 11.8% for random and systematic errors, respectively, for a total average error of 18.8% on the $NH_3$ total columns. The mean degrees of freedom for signal (DOFS) averaged over the 2002-2018 time series was 1.10.

## 2.2 In-Situ Measurements

To complement the FTIR total column $NH_3$ measurements, the publicly available in-situ data obtained by Environment and Climate Change Canada (ECCC) as a part of the National Air Pollution Surveillance Program (NAPS) were used (http://maps-cartes.ec.gc.ca/rnspa-naps/data.aspx) (National Air Pollution Surveillance Program). The data span December 2013 to April 2017, with a sampling frequency of one in three days. The sampling interval is 24 hours, from 00:00 to 24:00 local time, and samples were collected with a Met One SuperSASS-Plus Sequential Speciation Sampler. The detection limit is 0.6 ppb (Yao and Zhang, 2013). The integrated samples were brought back to the lab for analysis (Yao and Zhang, 2016). While errors are not reported in the dataset, the uncertainty is 10% when the $NH_3$ VMR is between 3 to 20 ppb (Hu et al., 2014). The instrument is situated less than 500 m away from the TAO FTIR, at 43.66°N, 79.40°W, 63 masl.

## 2.3 IASI Measurements

IASI is a nadir-viewing FTIR spectrometer on board the Metop-A, Metop-B and Metop-C polar-orbiting satellites, operated by the European Organization for the Exploitation of Meteorological Satellites (EUMETSAT), which have been operational since 2006, 2012 and 2018, respectively. IASI records spectra in the 645-2760 $cm^{-1}$ spectral range at a resolution of 0.5 $cm^{-1}$, with apodization. IASI can make off-nadir measurements up to 48.3° on either side of the track, leading to a swath of about 2×1100 km. At nadir, the field-of-view is $2 \times 2$ circular pixels, each at 12 km in diameter (Clerbaux et al., 2009).



The IASI NH$_3$ total columns (IASI ANNI-NH3-v3) are retrieved using an artificial neural network retrieval algorithm, with ERA5 meteorological reanalysis input data (Van Damme et al., 2017; Franco et al., 2018). Due to this retrieval scheme, there are no averaging kernels nor vertical sensitivity information for the retrieved columns (Van Damme et al., 2014). Details of the

retrieval scheme and error analysis can be found in Whitburn et al. (2016) and Van Damme et al. (2017). IASI-A and IASI-B NH$_3$ were combined and used in this study, as this allows for a longer time series and more data points for robust analysis. The retrieved columns of NH$_3$ from both satellites have been shown to be consistent with each other (Clarisse et al., 2019b; Viatte et al., 2020).

## 2.4 GEOS-Chem

The GEOS-Chem (v11-01) global chemical transport model (CTM) (geos-chem.org) was used in this study to supplement and compare against observational data. The model was run at $2° \times 2.5°$ resolution (latitude $\times$ longitude) using MERRA2 (Modern-Era Retrospective analysis for Research and Applications, Version 2) meteorological fields (Molod et al., 2015), the EDGAR emissions database (Janssens-Maenhout et al., 2019) for anthropogenic emissions, and NH$_3$ emissions (natural and anthropogenic) provided by Bouwman et al. (1997) and Croft et al. (2016). The GEOS-Chem model includes a detailed

tropospheric oxidant chemistry, as well as aerosol simulation (e.g., $H_2SO_4$-$HNO_3$-$NH_3$ simulation) (Park et al., 2004). NH$_3$ gas-aerosol partitioning is calculated using the ISORROPIA II model (Fountoukis and Nenes, 2007). Chemistry and transport are calculated with 20 and 10 minute timesteps, respectively. The model was spun up for one year, and output was saved every hour.

## 2.5 TAO and IASI Comparison

To assess the representative spatial and temporal scale of TAO NH$_3$ columns, the NH$_3$ total column measurements around Toronto made by IASI were compared against TAO total columns and modeled NH$_3$ columns (see Section 3.3). As NH$_3$ shows high spatiotemporal variability, several definitions of coincident measurements were used in this study, with spatiotemporal criteria of varying strictness. As NH$_3$ concentrations can vary significantly during the day, the temporal coincidence criterion was chosen to be $\leq 90$ minutes (Dammers et al., 2016). In addition, values of $\leq 60, 45, 30$ and 20 minutes were also tested.

For spatial coincidence criteria, $\leq 25$ km (Dammers et al., 2016), 30 km, 50 km, and 100 km were tested. For each criterion, correlations (both $r$ and slope) were calculated. This analysis was used to evaluate the spatial and temporal scales represented by the TAO NH$_3$ columns.

## 2.6 Trend Analysis and Identifying Pollution Events

With 16 years of data, relatively long-term trends of TAO FTIR column time series can be examined. While a trend analysis

simply using monthly averages is possible (Angelbratt et al., 2011), a more sophisticated method of fitting Fourier series of several orders was utilized in this study (Weatherhead et al., 1998). Bootstrap resampling was utilized to derive the confidence interval of the trends (Gardiner et al., 2008). A $Q$ value (the number of bootstrap resampling ensemble members generated for





statistical analysis) of 5000 was used (Gardiner et al., 2008). An additional analysis to determine the number of years of measurements needed to give the derived trend statistical significance ($2\sigma$ confidence) was also conducted, following Weatherhead

et al. (1998). This analysis takes into account the need for longer time series to identify trends in data that are autocorrelated (as are atmospheric observations). It should be noted that a major limitation of this analysis is that it assumes that data are collected at regular intervals, while TAO measurements are made at irregular intervals (due to the need for sunny conditions). For this reason, the confidence intervals derived from bootstrap resampling is a more robust method of error analysis, in the case of TAO data. However, as pointed out by Weatherhead et al. (1998), failing to take into account autocorrelation of the noise

can lead to underestimations of actual uncertainty, and for this reason, both bootstrap resampling and the Weatherhead method were used in this study. These techniques were combined to assess the intra- and inter-annual trends of $NH_3$ derived from TAO measurements, including a linear trend of the $NH_3$ total column along with its uncertainties and statistical significance. This analysis was also applied to the NAPS in-situ and IASI data.

The Fourier fit was used to identify $NH_3$ enhancements, following Zellweger et al. (2009). This analysis is done by taking

the negative residuals of the fit (i.e., measured values smaller than fitted values), mirroring them, and calculating the standard deviation ($\sigma$) of the mirrored residuals. Any measurements that are $2\sigma$ above the fit are considered enhancements. This analysis reduces biases in the spread due to enhancements by mirroring the negative residuals.

In this study, Fourier series of order 3 were utilized for all analyses. An analysis was done by comparing Fourier series fits of order 1 to 7, and checking for overfitting by running the residuals of the fit through a normality test (the Kolmogorov-Smirnov

test). While overfitting was not observed at higher orders, higher orders did not give more statistically-significant trends, so order 3 was chosen.

## 3  Results and Discussion

### 3.1  FTIR Measurements

The FTIR total column time series of $NH_3$ is shown in Figure 1. The purple points indicate enhancements, and the trends

(with and without outliers) are shown as red and cyan lines, respectively. The trend from 2002 to 2018 was found to be $3.34 \pm 0.46$ %/year and $2.23 \pm 0.79$ %/year ($2\sigma$ confidence interval from bootstrap resampling), with and without outliers, respectively (see Table 2). The number of years of measurements needed for the trend to be statistically ($2\sigma$) significant was found to be 33.8 years and 29.3 years, with and without enhancement events, respectively. Due to the irregular FTIR measurement intervals, these numbers may not represent the true significance of the trends, and should be regarded as best estimates of the significance

of the observed trend. The lower magnitude of the upward trend in the analysis without enhancement values indicates that the intra-annual variability of $NH_3$ is increasing. This is also evident when comparing the mean total column and standard deviations from, for example, the periods 2002-2005 and 2015-2018. In the former period, the mean $NH_3$ total column and standard deviation ($1\sigma$) were $5.94 \pm 5.14 \times 10^{15}$ molecules/cm$^2$, while in the latter time frame, they were $8.13 \pm 7.88 \times 10^{15}$ molecules/cm$^2$. The observed trend at TAO is comparable to a study by Warner et al. (2017), who observed an increasing $NH_3$

trend of 2.61 %/year over the United States from 2002 to 2016 using data from the Atmospheric Infrared Sounder (AIRS)





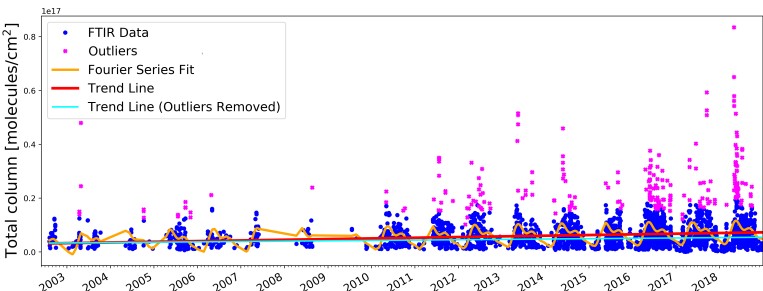

**Figure 1.** Time series of TAO FTIR NH$_3$ total columns from 2002 to 2018 with third-order Fourier series fit and linear trends.

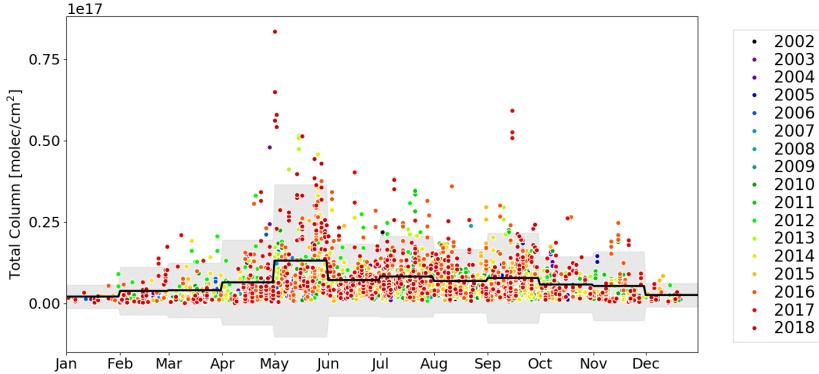

**Figure 2.** TAO FTIR NH$_3$ total columns plotted from January to December for 2002 to 2018. Monthly averages and $\pm\,2\sigma$ are indicated by the black line and shading, respectively.

satellite-based instrument.

Figure 2 shows the annual cycle of the FTIR NH$_3$ total columns, color coded by year, along with the monthly averages and $\pm$ $2\sigma$. TAO NH$_3$ columns have a maximum in May with a monthly total column average of $13.14 \pm 11.69 \times 10^{15}$ molecules/cm$^2$, due to agricultural emissions increasing in spring/summer (Hu et al., 2014; Dammers et al., 2016). The TAO seasonal cycle is consistent with findings by Van Damme et al. (2015b), who observed maximum NH$_3$ columns over the central United States during March-April-May (MAM). The mean NH$_3$ total column across the entire FTIR time series was $7.53 \pm 7.10 \times 10^{15}$ molecules/cm$^2$. These values are higher than remote areas, such as Eureka (located at 80.05°N, 86.42°W), where the highest monthly average was $0.279 \times 10^{15}$ molecules/cm$^2$, in July (Lutsch et al., 2016). However, TAO NH$_3$ total columns are far below values observed by the FTIR in Bremen (located at 53.10°N, 8.85°E), which saw values in the range of $\sim$100 $\times10^{15}$ molecules/cm$^2$ (Dammers et al., 2016). Monthly mean NH$_3$ columns are listed in Table 1.





**Table 1.** Monthly mean ($1\sigma$ in parenthesis) $NH_3$ total columns at TAO (2002-2018).

| Month | Mean Columns ($\times 10^{15}$ molecules/cm$^2$) |
|---|---|
| January | 2.11 (1.81) |
| February | 3.84 (3.68) |
| March | 4.05 (4.19) |
| April | 6.48 (6.52) |
| May | 13.14 (11.69) |
| June | 7.17 (5.54) |
| July | 8.27 (6.19) |
| August | 6.89 (4.89) |
| September | 7.81 (6.91) |
| October | 5.81 (4.30) |
| November | 5.36 (5.28) |
| December | 2.59 (1.81) |
| Overall Mean | 7.53 (7.10) |

**Table 2.** Comparison of $NH_3$ trends and $2\sigma$ confidence intervals observed in Toronto. All trends are in %/year.

| Dataset | Timeframe | Trends | Trends without outliers | TAO trends during the same timeframe |
|---|---|---|---|---|
| TAO | 2002-2018 | $3.34 \pm 0.46$ | $2.23 \pm 0.79$ | - |
| NAPS | 2013-2017 | $8.88 \pm 2.83$ | $6.40 \pm 0.18$ | $9.31 \pm 2.86$ |
| IASI | 2008-2018 | $8.38 \pm 0.77$ | - | $4.02 \pm 0.74$ |

## 3.2 NAPS Measurements

The NAPS in-situ $NH_3$ time series is shown in Figure 3. The purple points indicate enhancements, and trendlines with and without these outliers are shown as the red and cyan lines, respectively. The trendline was found to have a slope of $8.88 \pm 2.83$ %/year and $6.40 \pm 0.18$ %/year ($2\sigma$ confidence interval from bootstrap resampling), with and without outliers, respectively. The number of years needed for this trend to be $2\sigma$ significant was 8.4 years for both. Since NAPS data have very regular measurement intervals (once every three days), they are well suited for this trend significance analysis. Given that NAPS data only spans 3 years and 5 months, and 8.4 years of measurements are needed for $2\sigma$ confidence in the observed trend, it is uncertain if the increase in $NH_3$ levels is a definitive trend. For comparison, analysis of TAO $NH_3$ total columns during the same time period resulted in trends of $9.31 \pm 2.86$ %/year and $7.42 \pm 0.38$ %/year, with and without outliers, respectively.

The in-situ $NH_3$ VMRs were compared against the TAO FTIR columns by standardizing both measurements, following





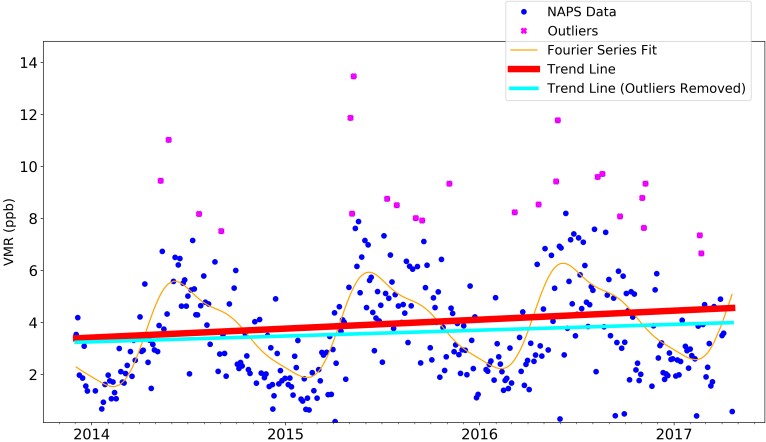

**Figure 3.** Time series of NAPS NH$_3$ surface VMR from 2013 to 2017 with third-order Fourier series fit and linear trends.

Equation 1 of Viatte et al. (2020):

$$X^i_{\text{standardized}} = \frac{X_i - \mu_X}{\sigma_X} \tag{1}$$

where $X$ is the dataset, indexed by $i$, $\mu$ is the mean, and $\sigma$ is the standard deviation of the dataset. The standardized dataset is centered around zero, and normalized by the standard deviation of the measurements. As the standardized dataset is unitless, it allows for comparison between different measurements in different units. In this study, the TAO NH$_3$ total columns were used, because the DOFS for the retrieval was around 1 (mean DOFS of the entire time series was 1.10), meaning there is only about one piece of vertical information in these measurements.

Standardized TAO and NAPS NH$_3$ are plotted in Figure 4a. Monthly averages and monthly standard deviations are shown in Figure 4b. The two measurements show similar seasonal cycles, with a maximum in May, and a minimum in December and January. There is a smaller secondary peak in November for both measurements. This may be due to late-season fertilizer application and cover crop growth. The Ministry of Agriculture, Food and Rural Affairs of Ontario recommends applying fertilizer in spring and fall (Munroe et al., 2018). Correlation between the two datasets can be seen in Figure 5a, where each NAPS measurement is plotted against the average of TAO measurements on that day (if any measurements are available). This simple comparison does not show a strong correlation, with $r = 0.51$, and slope $= 0.501$. However, resampling the measurements by 15-day averages (Figure 5b), 18-day averages (Figure 5c), 24-day averages (Figure 5d) and by monthly averages (Figure 5e), show much stronger correlations. Resampling to 15-day averages show better correlation with $r = 0.63$, and a larger slope $= 0.707$. Averaging to every 18 days and 24 days leads to $r = 0.68$ and 0.72, respectively. Monthly averages show the highest correlation with $r = 0.75$, and a slope $= 0.758$. This indicates that TAO and NAPS see similar low-frequency variabilities (period of 2 weeks or longer) in NH$_3$.





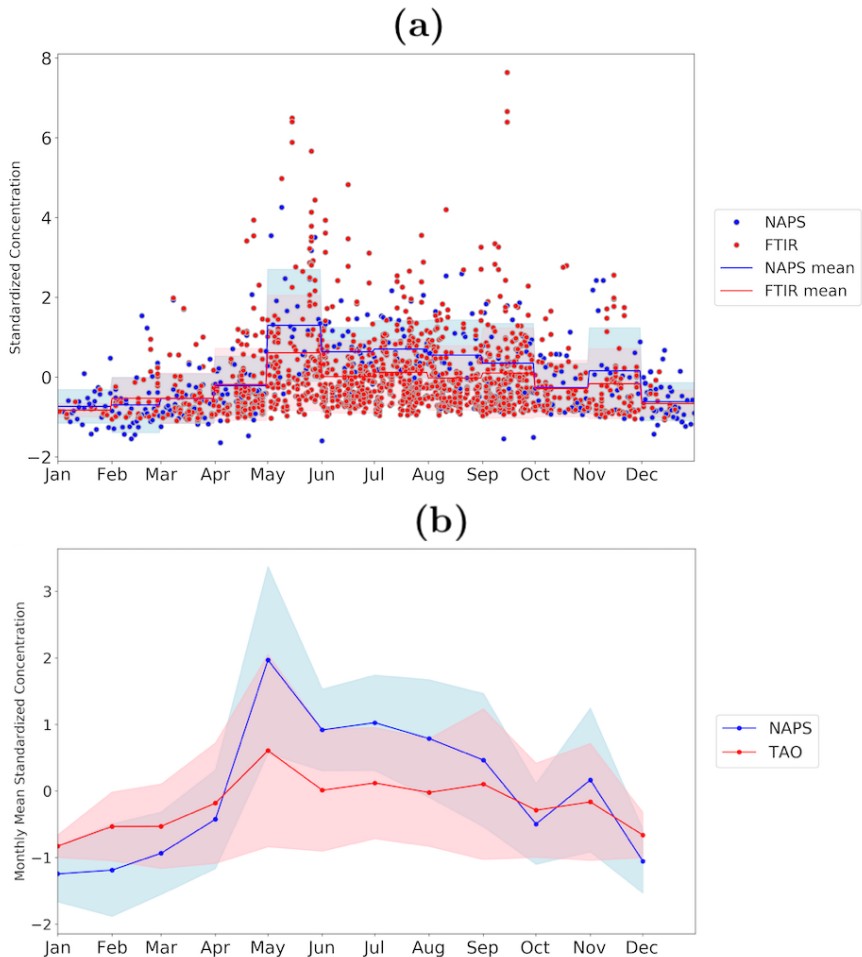

**Figure 4.** (a) Standardized TAO total column and NAPS surface VMR of $NH_3$ plotted from January to December. Monthly averages and $\pm$ $1\sigma$ are indicated by the red and blue lines and shading for TAO and NAPS, respectively. (b) The standardized TAO $NH_3$ total column (red) and NAPS surface $NH_3$ VMR (blue) monthly averages lines with their respective $\pm\ 1\sigma$ (shading).

### 3.3 IASI Measurements

The time series of IASI $NH_3$ total columns (2008 to 2018) within 50 km of TAO is shown in Figure 6. The trend of these IASI measurements is $8.38 \pm 0.77$ %/year, where the error indicates the $2\sigma$ confidence interval obtained by bootstrap resampling analysis. The Weatherhead et al. (1998) method for finding the statistical significance of this trend was not utilized here, as the analysis requires calculating the autocorrelation of data, which is not possible given the spatially scattered dataset. For comparison, the TAO FTIR trend over the same period is $4.02 \pm 0.74$ %/year.

The correlations between IASI and TAO $NH_3$ columns for the various coincidence criteria listed in Section 2.3 are shown in



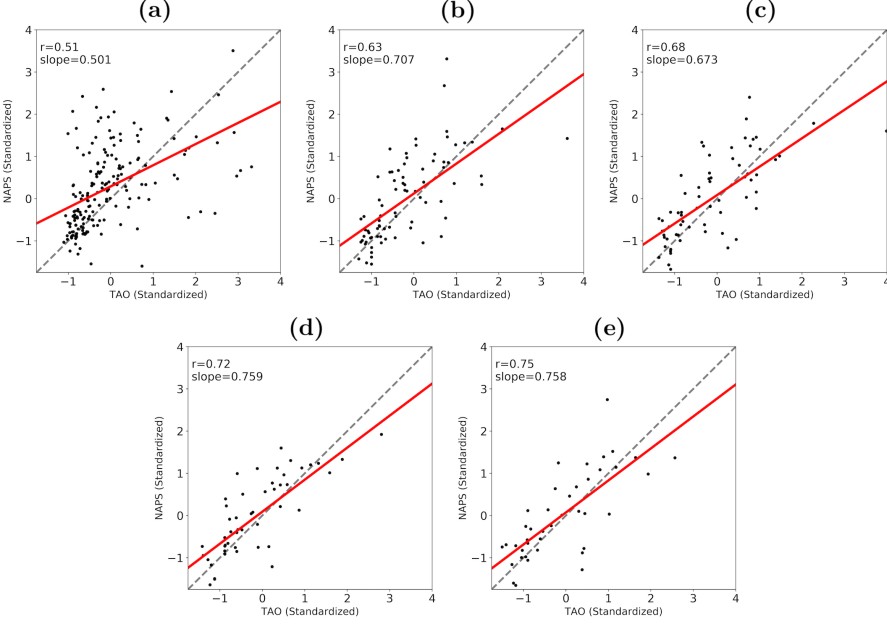

**Figure 5.** Standardized NAPS $NH_3$ surface VMR plotted against standardized TAO FTIR $NH_3$ total column. (a) The raw comparison, where for each NAPS observation, the closest daily average TAO measurement, if there are any within 72 hours, is plotted. TAO and in-situ resampled to (b) 15-day, (c) 18-day, (d) 24-day and (e) monthly averages. The dashed lines indicate slope = 1, and the red lines indicate the fit to the data.

Table 3, along with the slopes, mean relative difference (MRD), and total number of data points. The MRD was calculated by subtracting the TAO column from the IASI column, then dividing by the TAO column (Dammers et al., 2016). To maximize the number of coincident data points, no significant data filtering (e.g., filtering by relative errors) was performed. The criteria used by Dammers et al. (2016) (90 minutes, 25 km) shows a correlation with $r = 0.65$ and slope = 0.88 in this study, comparable to $r = 0.79$ and slope = 0.84 reported by Dammers et al. (2016). The MRD was $-45.5 \pm 207.2$ % for this study, consistent with $-46.0 \pm 47.0$ % calculated by Dammers et al. (2016) for TAO data. The larger standard deviation of the MRD is most likely because the data used here were not filtered by relative errors. The best correlation was achieved when using measurements made within 20 minutes and within 25 km of each other, which resulted in $r = 0.73$ and slope of 1.14. Coincidence criteria of 20 minutes and 50 km gave $r = 0.68$ and slope = 1.06. Criteria of 45 minutes and 50 km also shows a correlation comparable to the 90 minutes, 25 km criteria, with $r = 0.64$ and slope = 0.92. This suggests that TAO FTIR is a good indicator of $NH_3$ concentrations on a city-wide scale ($\sim 50$ km). This is also evident when looking at the correlation between TAO columns vs. daily averaged IASI measurements within 50 km, which had $r = 0.69$, although the slope was smaller, at 0.82. The better correlations seen with the stricter temporal criteria suggest that $NH_3$ near Toronto exhibits high-frequency variability. The values obtained in this study are also comparable to recent findings by Tournadre et al. (2020), who compared $NH_3$ columns from an FTIR stationed in Paris to IASI $NH_3$ columns. With a 15 km and 30 minutes coincidence criteria, the FTIR in Paris








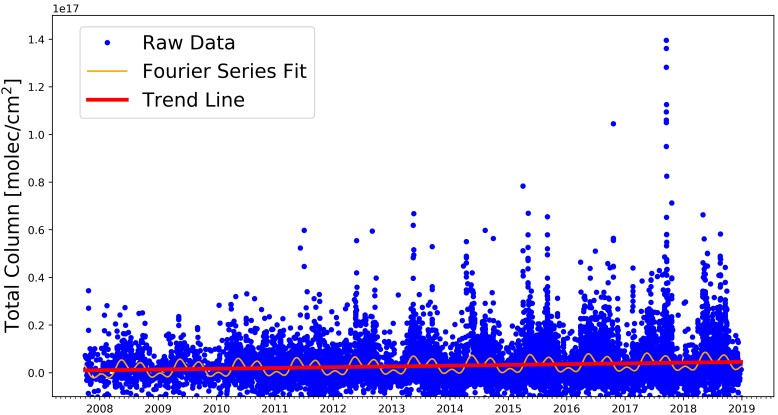

**Figure 6.** Time series of IASI NH$_3$ total columns measured within 50 km of TAO. The third-order Fourier series fit and the trend line are shown in orange and red, respectively.

showed a correlation of $r = 0.79$ and slope = 0.73. The same study also found that the FTIR in Paris is capable of providing information about NH$_3$ variability at a "regional" scale ($\sim$ 120 km) Tournadre et al. (2020). Although not quantified in this study, the line-of-sight through the atmosphere (which changes throughout the day) may also affect the representative scale of

ground-based solar-pointing FTIR observations. Additionally, the number of observations is relatively large for each criterion (e.g., $N = 923$ for 90 minutes, 25 km, while $N = 679$ for 45 minutes, 50 km), suggesting that the differences in correlation are not simply due to the differences in the number of data points. The correlation plots for 20 min/25 km, 90 min/25 km, 20 min/50 km and 45 min/50 km are shown in Figures 7a, 7b, 7c and 7d, respectively. It should be noted that the slope was calculated through a simple linear regression. For comparison, an additional analysis was done propagating measurement uncertainty

using the unified least squares procedure outlined by York et al. (2004) and yielded similar results, with a smaller slope for all cases due to the larger relative uncertainty on IASI measurements ($\sim$68 % for IASI compared to $\sim$19 % for TAO).

IASI column and NAPS surface NH$_3$ were also compared in this study by converting to standardized data (see Equation 1). Comparing the monthly means resulted in $r = 0.79$ and slope = 0.79 when looking at IASI measurements made within 50 km of NAPS, and $r = 0.74$ and slope = 0.74 for 30 km. Without temporal resampling, no significant correlation was found ($r \leq 0.27$)

for any spatial coincidence criteria. This is in line with findings from Van Damme et al. (2015a), where significant correlation was found when comparing monthly averaged surface and IASI measurements. Van Damme et al. (2015a) report $r = 0.28$ when comparing IASI with an ensemble of surface observations over Europe, and $r$ as high as 0.81 and 0.71 for measurements made at Fyodorovskoye, Russia and the Monte Bondone, Italy, respectively. It should be noted that the comparisons in Van Damme et al. (2015a) were done by converting IASI NH$_3$ columns to surface concentration by using the same model used in the

retrieval process, as opposed to the standardized dataset approach used in this study.





**Table 3.** IASI vs. TAO correlation coefficient, slope (regression standard error in parenthesis), MRD ($1\sigma$ RMS in parenthesis) (in %), and number of data points, calculated for each TAO measurement, for varying spatial and temporal coincidence criteria.

| Coincidence criteria | ≤ 25 km | ≤ 30 km | ≤ 50 km | ≤ 100 km |
|---|---|---|---|---|
| ≤ 20 minutes | r = 0.73<br>slope = 1.14 (0.06)<br>MRD = -61.9 (161.6) %<br>N = 314 | r = 0.72<br>slope = 1.11 (0.06)<br>MRD = -58.3 (156.7) %<br>N = 337 | r = 0.68<br>slope = 1.06 (0.06)<br>MRD = -51.2 (166.1) %<br>N = 384 | r = 0.63<br>slope = 1.24 (0.08)<br>MRD = -47.7 (190.9) %<br>N = 421 |
| ≤ 30 minutes | r = 0.71<br>slope = 1.06 (0.05)<br>MRD = -48.1 (216.8) %<br>N = 438 | r = 0.70<br>slope = 1.04 (0.05)<br>MRD = -43.1 (204.2) %<br>N = 470 | r = 0.65<br>slope = 0.98 (0.05)<br>MRD = -42.0 (200.4) %<br>N = 528 | r = 0.59<br>slope = 1.07 (0.06)<br>MRD = -40.9 (186.7) %<br>N = 575 |
| ≤ 45 minutes | r = 0.68<br>slope = 0.93 (0.04)<br>MRD = -47.4 (198.2) %<br>N = 588 | r = 0.67<br>slope = 0.92 (0.04)<br>MRD = -42.8 (190.6) %<br>N = 623 | r = 0.64<br>slope = 0.92 (0.04)<br>MRD = -41.0 (185.5) %<br>N = 679 | r = 0.58<br>slope = 0.97 (0.05)<br>MRD = -42.4 (177.9) %<br>N = 732 |
| ≤ 60 minutes | r = 0.66<br>slope = 0.89 (0.04)<br>MRD = -46.4 (188.4) %<br>N = 708 | r = 0.65<br>slope = 0.90 (0.04)<br>MRD = -42.8 (180.3) %<br>N = 750 | r = 0.62<br>slope = 0.89 (0.04)<br>MRD = -38.2 (176.7) %<br>N = 815 | r = 0.56<br>slope = 0.93 (0.05)<br>MRD = -40.7 (172.1) %<br>N = 866 |
| ≤ 90 minutes | r = 0.65<br>slope = 0.88 (0.03)<br>MRD = -45.5 (207.2) %<br>N = 923 | r = 0.65<br>slope = 0.88 (0.03)<br>MRD = -44.1 (192.4) %<br>N = 967 | r = 0.61<br>slope = 0.89 (0.04)<br>MRD = -40.7 (193.5) %<br>N = 1039 | r = 0.56<br>slope = 0.93 (0.04)<br>MRD = -36.9 (186.3) %<br>N = 1093 |

## 4 Comparison with GEOS-Chem

The $NH_3$ total column from the GEOS-Chem CTM model grid cell containing Toronto (grid center at 44°N, 80°W) is shown in Figure 8a, along with TAO FTIR data. The correlation was obtained by comparing the hourly model data for each FTIR observation. Comparison with the FTIR was done with and without smoothing the model data with the FTIR averaging kernel and a priori profile (Rodgers and Connor, 2003). As smoothing the model data only resulted in differences of less than 1%, the discussion here will focus on the unsmoothed dataset to be consistent with the comparison with IASI. While GEOS-Chem is able to capture the seasonal cycle seen at TAO, the correlation is not strong, with $r = 0.51$ and the coefficient of determination, $r^2$, at 0.26 (see Figure 9a). The calculated slope was 1.16. Both of these values are without smoothing the model data. Smoothing the data resulted in $r^2 = 0.28$, and slope = 1.01 It is likely that the model is too coarse (the 2° × 2.5° grid box corresponds to approximately 220 km × 200 km), and TAO, while able to capture larger-scale variability in $NH_3$ than in-situ observations, is not sensitive to observations at spatial scales of 100 km or larger. Given the short lifetime of $NH_3$, it is



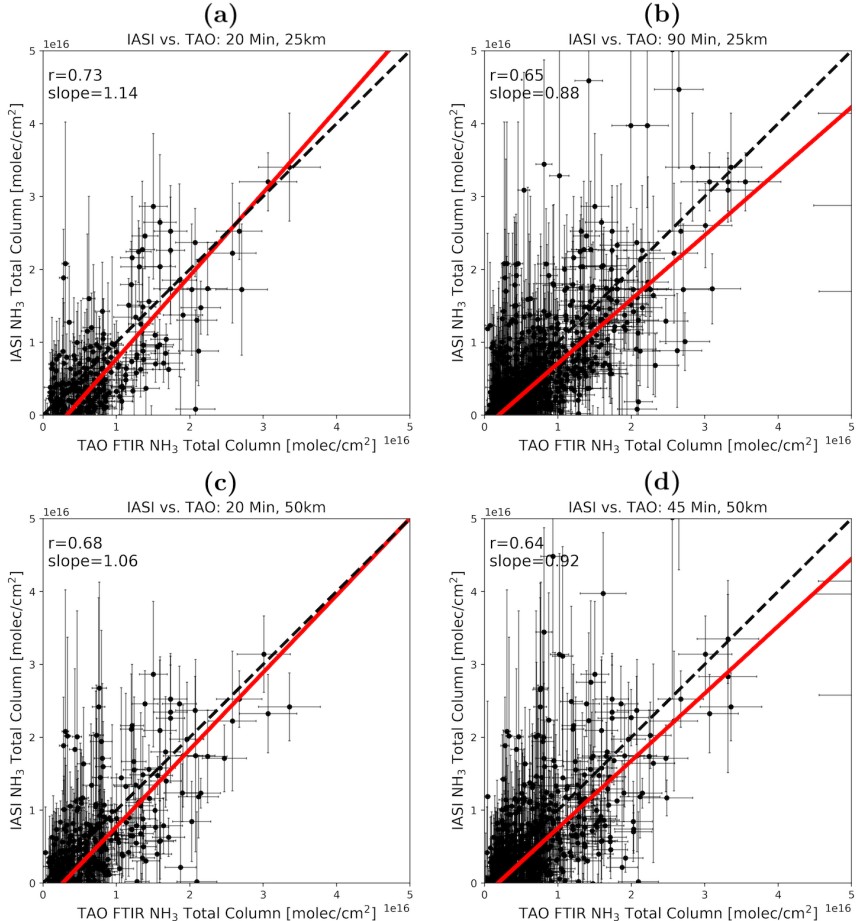

**Figure 7.** Correlation plots for IASI vs. TAO FTIR NH$_3$ total columns, with coincidence criteria of (a) 20 minutes and 25 km, (b) 90 minutes and 25 km, (c) 20 minutes and 50 km, and (d) 45 minutes and 50 km. Data from 2008 to 2018 are plotted. Dashed lines indicate slope = 1, while the red lines are the lines of best fit. Error bars are the reported observational uncertainties.

unsurprising to see large variability at these spatial scales.

For comparison with IASI, a larger domain was chosen to assess the correlation of the model and satellite observations at a larger regional scale. Model grids spanning 35°N to 53°N, and 93.75°W to 63.75°W were used for the analysis, as these 280 grids capture Toronto, the Great Lakes, and the Atlantic Ocean coastline. The spatial coincidence was calculated by binning the IASI data into the grids of GEOS-Chem, and temporal coincidence was determined by calculating the mean overpass time in the domain and averaging the model data between one hour before and one hour after the mean overpass time. The time series (both GEOS-Chem and IASI were averaged over the domain) and correlation plots are shown in Figures 8b and 9b, respectively. Correlation of GEOS-Chem against IASI is higher than GEOS-Chem against TAO FTIR, with $r^2 = 0.33$. This is





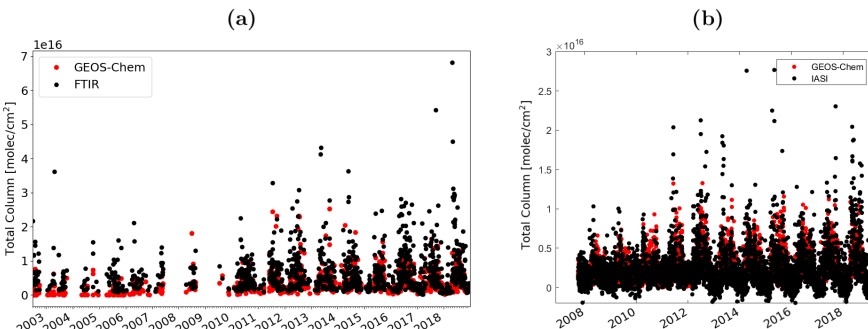

**Figure 8.** (a) GEOS-Chem and TAO FTIR NH$_3$ total columns from 2002 to 2018. GEOS-Chem data shown here were not smoothed with the FTIR averaging kernel and a priori profile. (b) GEOS-Chem and IASI NH$_3$ total columns (averaged over domain spanning from 35°N to 53°N, and 93.75°W to 63.75°W) from 2008 to 2018.

comparable to findings by Schiferl et al. (2016), who observed IASI and GEOS-Chem correlations ($r$) of 0.6–0.8 in the United States Great Plains and the Midwest during the summer. The slope was 0.85, meaning NH$_3$ is overestimated in GEOS-Chem when compared to IASI at this scale. Comparing GEOS-Chem and IASI for one grid cell over Toronto (same cell as the one used for comparison with TAO FTIR) resulted in a lower correlation, at $r^2 = 0.13$. These results suggest GEOS-Chem is able to model NH$_3$ on larger regional scales, but a finer resolution is needed for better comparison with smaller regions. In addition,

while the modeled NH$_3$ was overestimated in comparison with IASI over a larger regional domain, the comparison for the single grid box over Toronto resulted in slope = 2.44, meaning the model underestimated NH$_3$ in this smaller region, which may indicate underestimation of local NH$_3$ sources near Toronto in the model. This can be contrasted to recent findings by Van Damme et al. (2014), who observed an overall underestimation of NH$_3$ in the LOTOS-EUROS model over Europe when compared against IASI. In a four-year period from 2008 to 2011 over the Netherlands, for example, IASI NH$_3$ columns are as

high as 6.5 mg/m$^2$, while the modeled NH$_3$ go up to 5.2 mg/m$^2$ (Van Damme et al., 2014).

## 5   Conclusions

     The TAO FTIR spectrometer situated in downtown Toronto, Ontario, Canada has been used to obtain a 16-year time series of total columns of NH$_3$. These columns were compared against other NH$_3$ observations (IASI column and NAPS in-situ surface VMR) and GEOS-Chem model data. Analysis of TAO NH$_3$ columns showed an upward annual trend of 3.34 ± 0.46 % and

2.23 ± 0.79 % over the period 2002-2018, with and without outliers, respectively. The larger trend with outliers included suggests that NH$_3$ enhancements are becoming more frequent and seasonal variability is increasing. These values are in agreement with trends observed by other studies. For example, Warner et al. (2017) observed a trend of 2.61 %/year from 2002 to 2016 over the USA using data from the Atmospheric Infrared Sounder (AIRS) aboard NASA's Aqua satellite, and Yu et al. (2018) derived surface NH$_3$ trends of ∼5 % and ∼5-12 % in the Western and Eastern United States from 2001 to 2016, respectively,



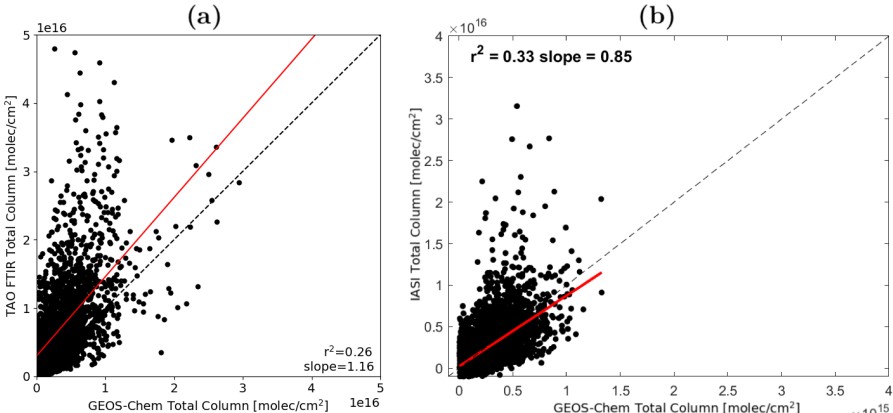

**Figure 9.** Correlation plots of (a) TAO vs. GEOS-Chem NH$_3$ total columns, and (b) IASI vs. GEOS-Chem NH$_3$ total columns. Data from 2002 to 2018 are plotted for TAO and data from 2008 to 2018 are plotted for IASI.

using GEOS-Chem modeled NH$_3$.

Similar analysis of the NAPS in-situ time series showed that NH$_3$ at the surface is also increasing, with an annual increase of 8.88 ± 2.83 % and 6.40 ± 0.18 % calculated with and without outliers, respectively. TAO total columns during the same period showed trends of 9.31 ± 2.86 %/year and 7.42 ± 0.38 %/year with and without outliers, respectively. TAO FTIR and NAPS comparisons showed that TAO columns are well correlated with surface NH$_3$ when resampled to monthly means to

reduce high-frequency variability.

IASI NH$_3$ total columns measured within 50 km of TAO exhibited an annual trend of 8.38 ± 0.77 %/year from 2008 to 2018. For comparison, TAO FTIR NH$_3$ total columns over the same period showed a trend of 4.02 ± 0.74 %/year. The IASI columns were also compared against FTIR columns, with the good correlations being obtained with distance criterion of ∼50 km, indicating that the TAO FTIR measurements are representative of NH$_3$ at a city-size scale. Comparing different coinci-

dence criteria showed that, at least in Toronto, distance criteria can be larger than the 25 km used by Dammers et al. (2016), but temporal criteria may need to be stricter, at around ∼45 minutes (instead of 90 minutes). The highest correlation ($r = 0.73$) was seen with coincidence criteria of 25 km and 20 minutes.

TAO FTIR and IASI NH$_3$ columns were also compared with GEOS-Chem model data. The model did not show a very high correlation with TAO for a single grid cell containing Toronto, with $r^2 = 0.26$, and $r^2 = 0.28$ when the model data was

smoothed with the FTIR averaging kernel. The model comparison with IASI showed slightly better agreement on a domain spanning 35°N to 53°N, and 93.75°W to 63.75°W, with $r^2 = 0.33$. These results suggest that TAO, representative of NH$_3$ at a city-size scale (∼50 km), requires higher-resolution model runs for comparison. This is also evident when comparing GEOS-Chem against IASI within the single model grid cell that includes TAO; this comparison led to a poorer correlation with $r^2 = 0.13$. In addition, GEOS-Chem overestimated NH$_3$ in the larger domain when compared with IASI. However, in the single grid

cell over TAO, the model underestimated NH$_3$ columns compared to both IASI and TAO.



This study showed a positive trend of $NH_3$ over Toronto derived from ground-based FTIR, satellite, and in-situ measurements. The $NH_3$ total columns using an FTIR situated in downtown Toronto showed an observational footprint at a city-size scale, although this also highlights the need for models simulating $NH_3$ to be run at higher resolution than $2° \times 2.5°$ for comparisons with ground-based measurements.

*Data availability.* The TAO FTIR data used in this study are publicly available from the NDACC data repository hosted by NOAA at ftp://ftp.cpc.ncep.noaa.gov/ndacc/station/toronto/hdf/ftir/. The IASI ANNI-NH3-v3 data used in this study are available on request (M. Van Damme, ULB). The IASI Level-1C data are distributed in near real time by Eumetsat through the EumetCast system distribution. IASI Level-1C data and Level-2 $NH_3$ data can be accessed via the Aeris data infrastructure (http://iasi.aeris-data.fr/NH3/). The in-situ $NH_3$ data measured by the Canadian National Air Pollution Surveillance (NAPS) are available at http://data.ec.gc.ca/data/air/monitor/
national-air-pollution-surveillance-naps-program/, provided by Environment and Climate Change Canada (last access: November 2019) (National Air Pollution Surveillance Program). The GEOS-Chem model (v11-01) is freely available to the public. Latest model information can be found at The International GEOS-Chem User Community (2020). Instructions for downloading and running the models can be found at http://wiki.geos-chem.org/.

*Author contributions.* SY, CV and KS conceived this study. SY wrote the paper with contributions from all authors. EL and SY performed
the retrieval of $NH_3$ columns at TAO. SY ran the GEOS-Chem model with guidance from DBAJ. SY and CV performed the analyses and comparisons of $NH_3$ measurements around Toronto. MVD, LC, and SW performed IASI retrievals, and CV analyzed the IASI retrievals with guidance from CC and PFC. All of the authors discussed the results and contributed to the final paper.

*Competing interests.* No competing interests are present.

*Acknowledgements.* This project was made possible thanks to the 2018 Centre National de la Recherche Scientifique – University of Toronto
Call for Joint Research Proposals, which provided the initial support for the AmmonAQ project. This work, including measurements made at TAO, was supported by Environment and Climate Change Canada (ECCC), the Natural Sciences and Engineering Research Council (NSERC), and the NSERC CREATE Training Program in Technologies for Exo-Planetary Science. IASI is a joint mission of EUMETSAT and the Centre National d'Etudes Spatiales (CNES, France). The authors acknowledge the Aeris data infrastructure (http://iasi.aeris-data.fr/NH3/) for providing access to the IASI data, as well as ECCC for the NAPS in-situ data. ULB has been supported by the Belgian
State Federal Office for Scientific, Technical and Cultural Affairs (Prodex arrangement IASI.FLOW). LC and MVD are respectively research associate and postdoctoral researcher with the Belgian F.R.S-FNRS.



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
