# Peer review of "Multiscale observations of NH3 around Toronto, Canada"

_Atmospheric Measurement Techniques, 2020_

## Referee Comment (RC1) · Anonymous Referee #1 · 29 Sep 2020

In this investigation, three independent observational data sets are used to characterize the variability and trends of ammonia over the Toronto area. The results present a positive trend in the NH3 concentration over the region with larger values in the Spring, confirming what has been published in previous studies. The large amount of data analyzed in this contribution and the comparisons performed among these rich data sets (in situ, column and satellite) provide, however, the most precise and confident resutls on NH3 in the Toronto area so far. The comparisons of the column and satellite observations with CTM simulations on a coarse grid indicate, as expected, that the model is not properly capturing local sources responsable for the high frequency variability.

MAYOR

The approach to determine the observational footprint of the FTIR column measurements seems to be oversimplified. It is only based in correlating the data with the satellite observations at different spatial and temporal scales. The best correlation and slope is obtained with the most strict criteria (25km/20 min). A proper footprint analysis would require to take the wind fields within the considered time period in consideration, which is not done. Although this simple analysis gives some indication of the representativeness of ground-based measurement, it should not be claimed in the text that a proper observational footprint assessment has been performed.

A bias would be expected to be observed between the FTIR and in situ data just because the FTIR only measures during sunny conditions. NAPS data is collected regularly every third day. Moreover, NH3 has a strong diurnal pattern that is not reported in this paper. While in situ data represents the average concentration within a 24 h period, FTIR data is available only during the day. The authors contrast the trends from the linear regressions from both data sets (TAO and NAPS) when outliers are and are not considered (L204). However, no mention or explanation is given for this source of bias given that NH3 concentrations are probably expected to peak during warmer days and warmer hours. It would be interesting to compare both data sets only for coincident measurement days and give a more comprehensive explanation of this additional source of bias.

It seems that the comparison of both TAO and IASI data sets with GEOS-Chem is challenging due to the coarse resolution of the model. It is shown from the comparison of the ground-based data with the satellite observations that NH3 presents high frequency variability in the region. It would then seem logical that the authors filter out the enhancements from the FTIR data, as done in the trend analysis, before correlating to the model data. The same could apply to IASI data since the enhancements observed within the large model domain are probably due to local emissions that are not well represented by the model. Figures 9a and b could then show the correlation and regression results as is, as well as from the filtered data sets.

MINOR

L28. The sentence is not accurate. The health impact of PM2.5 is strongly dependent on the chemical composition and the cited study does not take composition into account. In the context of this contribution, the PM containing ammonium salts are not the most hazardous and also those that contribute to smog are rather organic in nature. Please rephrase.

L41. Referring to NH3 being injected to the free troposphere, you may want to cite Hoepfner et al 2016 (www.atmos-chem-phys.net/16/14357/2016/)

L86. A citation or description for the camera and solar disk-fitting system of the solar tracker is missing.

L90 Should say "... microwindows in the ... and ... spectral regions."

L76. Was there any quality control and data filtering performed? Please describe. Same for the in situ data.

L110. No need to repeat (National Air Pollution Surveillance Program)

L117. Define the IASI acronym.

L121. May not be clear to the reader what a 2 x 2 circular pixel is. Maybe a matrix of 2 x 2 pixels?

L126. Indicate the overpass times of each satellite instrument

L155 What do "longer time series" refer to? The length considered in this contribution? Please specify.

L165. If mirroring a value is the same as taking its absolute value, the readers might be more familiar with the second terminology. It may also be wise to mention that the average of the mirrored residuals include the positive ones.

Fig4. Figure 4 b) seems redundant since no additional information is provided with respect to a).

Fig5. It would seem sufficient to show the correlation plots a) and e) in this figure, while keeping the results of the different resampling periods in the text (L219-223)

L300. A larger trend with outliers with respect to that obtained without them may not be conclusive when looking at the data availability of the TOA data series. Measurements seem to be performed more regulary in recent years so to me the increase in seasonal variability is more evident when comparing for example the standard deviations year to year.

---

## Referee Comment (RC2) · Anonymous Referee #2 · 6 Oct 2020

**Multiscale observations of NH3 around Toronto, Canada, Yamanouchi et al., AMT, 2020**

**General Description:**

The authors intercompare NH3 concentrations obtained as column densities from groundbased and space-based instruments, surface concentrations from a monitoring site, and 3D mixing ratios from the GEOS-Chem model in and around Toronto, Canada. These they use to determine long-term trends in NH3 concentrations and assess consistency across the data platforms. The manuscript is in general well written and easy to follow, but requires additional details about the model setup and information about sulfate and nitrate in and around Toronto for interpretation of the NH3 concentration trends. These and other comments are provided below.

**General Comments:**

Why was the nested version of GEOS-Chem over North America not used? It includes Toronto in the domain and is at finer resolution  $(0.25^{\circ} \times 0.3125^{\circ})$  than the global domain.

There is quite a lot of information relevant to model representation of  $NH_3$  that is missing in the model description section. These include the following: The inventories used in the model to represent US and Canadian SO2 and NOx sources that form sulfate and nitrate that influence  $NH_3$  uptake to aerosols. The version of EDGAR and whether this is the inventory that represents anthropogenic  $NH_3$  emissions over the domain of interest or whether it is a combination of EDGAR and GEIA (now quite outdated and only really used in the model to represent natural  $NH_3$  emissions). The base year of each inventory. Whether annual scaling factors are applied to any of the emissions that would have declined due to emission regulations (typically  $NO_x$  and  $SO_2$ ). Whether seasonal scaling factors are applied to  $NH_3$ emissions in the model.

The model also seems to be underutilised to provide context for the study region. The inventories could, for example, be used to assess the relative proportion of vehicular, agricultural, and natural emissions to total  $NH_3$  emissions and to determine the role of changes in sulfate and nitrate (due to emission regulations of  $SO_2$  and  $NO_x$  sources) on observed trends in  $NH_3$ .

What is the fit that is applied to the data to obtain the trends? And what is the determination of significance? It is stated in the text that "The number of years of measurements needed for the trend to be statistically  $(2\sigma)$  significant was found to be 33.8 years and 29.3 years" (p. 6, lines 177-178), but it is not clear why this is the case given that the  $2\sigma$  uncertainty is much less than the trend value. An explicit statement of what the authors use as a significance criterion might help avoid confusion.

The FTIR instrument and measurements are referred to in figures/tables/text as FTIR, TAO, or TAO FTIR. To avoid confusion, stick with one of these throughout.

**Specific Comments:**

p1, line 14: There is no context for the use of "resampling" in the abstract to be able to follow what this implies for the results obtained. What is being resampled? And why does it alter the correlation?

p2, line 38: Briefly elaborate on the link between  $NH_3$  concentrations and  $SO_2$  and  $NO_x$  emissions.

p2, line 39: "...as well as by reactions with acids in the atmosphere" sounds like it is happening in the gas phase. Make clear that this is a heterogeneous process.

p3, line 59: What is the NH3 source from greenery? Application of fertiliser to gardens and public spaces?

p4, line 97: What is the shape of the a priori profile used for the retrieval? How does it compare to that from GEOS-Chem?

p4, line 121: Odd to express the swath like this. Standard is as 2200 km.

p5, line 137-138: Say what model years are sampled after the one year spin up.

p5, line 145-147: This approach is reasonable and widespread, but what if the spatial extent is less than the spatial resolution of IASI (at best 12 km at nadir), as seems to be the case in this work?

Figure 2: Does the seasonality differ if the median is calculated for each month?

Figure 2: Consider showing the y-axis as 1e16 rather than 1e17.

p7, line 189: Why is the seasonality solely attributed to emissions? What about partitioning of NH3 to acidic aerosols? Is there any seasonality to this process?

Table 1: Is there a reason that this table is included if this information is already illustrated in Figure 2?

Table 2: The layout of the table is confusing, as the row labels correspond to specific time periods, but then the final column is labelled "during the same timeframe". What is this timeframe then? Why is the FTIR TAO trend for this same timeframe not given?

Figure 4: The lines in (a) are not easy to see. Consider making these thicker.

p12, line 248: Tournadre et al. (2020) is not cited correctly.

p12, line 254: What is "simple linear regression"? Ordinary least squares?

p12, line 259-260: It's not clear what this means: "Without temporal resampling, no significant correlation was found ( $r \le 0.27$ ) for any spatial coincidence criteria". What is this temporal resampling and why does it impact the correlation?

Table 3: The information as presented in this table is okay, but would have been more visually interesting and easier to identify patterns in the data if each variable (r, slope etc.) was illustrated on 2D colored grids.

p13, line 267: What does this gridbox include other than Toronto that might dilute or increase NH3 concentrations and affect the comparison?

Figure 7: It would be helpful to say in the caption or text what this is showing from Table 3.

Figure 8: It is not easy to discern the red and black points in panel (b).

Figure 9: Are units for GEOS-Chem in panel (b) correct?

---

## Author Comment (AC1) · 22 Nov 2020

10.5194/amt-2020-319 Revisions: Response to Reviewer 1

Shoma Yamanouchi et al.

General Revisions:

**We thank the two reviewers for their helpful comments, which have enabled us to improve the manuscript. The reviewers' comments are in regular font below and our responses are in bold font. Line numbers in the responses refer to the revised manuscript with changes tracked. Also of note is that there was a minor bug in the trend analysis code; this was revised, and affected values were corrected (this only affected the 2$\sigma$ confidence intervals from bootstrap resampling).**

Reviewer 1
The approach to determine the observational footprint of the FTIR column measurements seems to be oversimplified. It is only based in correlating the data with the satellite observations at different spatial and temporal scales. The best correlation and slope is obtained with the most strict criteria (25km/20 min). A proper footprint analysis would require to take the wind fields within the considered time period in consideration, which is not done. Although this simple analysis gives some indication of the representativeness of ground-based measurement, it should not be claimed in the text that a proper observational footprint assessment has been performed.

**All sentences claiming that the "footprint assessment" was performed were replaced by sentences that mention the representativeness of ground-based measurements.**

A bias would be expected to be observed between the FTIR and in situ data just because the FTIR only measures during sunny conditions. NAPS data is collected regularly every third day. Moreover, NH3 has a strong diurnal pattern that is not reported in this paper. While in situ data represents the average concentration within a 24 h period, FTIR data is available only during the day.

**A brief discussion of this bias was added (Line 248).**

The authors contrast the trends from the linear regressions from both data sets (TAO and NAPS) when outliers are and are not considered (L204). However, no mention or explanation is given for this source of bias given that NH3 concentrations are probably expected to peak during warmer days and warmer hours. It would be interesting to compare both data sets only for coincident

measurement days and give a more comprehensive explanation of this additional source of bias.

**The comparison analysis using only coincident measurements is shown in Figure 5a. A brief discussion on warmer days and higher $NH_3$ was added, along with an additional analysis to examine coincident FTIR and in-situ measurements and temperatures; on three occasions where simultaneous enhancements were observed in the FTIR and in-situ data (once in May 2014, twice in May 2016), the daily average temperatures were higher than the monthly averages (Line 248-259).**

It seems that the comparison of both TAO and IASI data sets with GEOS-Chem is challenging due to the coarse resolution of the model. It is shown from the comparison of the ground-based data with the satellite observations that NH3 presents high frequency variability in the region. It would then seem logical that the authors filter out the enhancements from the FTIR data, as done in the trend analysis, before correlating to the model data. The same could apply to IASI data since the enhancements observed within the large model domain are probably due to local emissions that are not well represented by the model. Figures 9a and b could then show the correlation and regression results as is, as well as from the filtered data sets.

**This analysis was performed. Filtered FTIR measurements compared with GEOS-Chem resulted in $r^2 = 0.22$ and slope = 0.68 (when no filtering was performed, the values were $r^2 = 0.26$ and slope = 1.16). Comparisons of filtered IASI observations and GEOS-Chem resulted in $r^2 = 0.29$ and slope = 0.57 (when no filtering was performed, the values were $r^2 = 0.33$ and slope = 0.85). Corresponding plots were added (Figures 9c and 9d).**

L28. The sentence is not accurate. The health impact of PM2.5 is strongly dependent on the chemical composition and the cited study does not take composition into account. In the context of this contribution, the PM containing ammonium salts are not the most hazardous and also those that contribute to smog are rather organic in nature. Please rephrase.

**The sentence was reworded and another reference added here (Schiferl et al., 2014). We are not claiming that particulate matter forming due to ammonium salts are the most hazardous. Additionally, recent studies (e.g., Liu et al., 2019; Wielgosiński & Czerwińska, 2020) have shown that ammonium salts do contribute to smog as well as haze. The sentence was also reworded clarify this (Line 29-34).**

L41. Referring to NH3 being injected to the free troposphere, you may want to cite Hoepfner et al 2016 ([www.atmos-chem-phys.net/16/14357/2016/](www.atmos-chem-phys.net/16/14357/2016/))

**The reference was added (Line 48).**

L86. A citation or description for the camera and solar disk-fitting system of the solar tracker is missing.

**Further details can be found in Franklin (2015, [http://hdl.handle.net/10222/64642](http://hdl.handle.net/10222/64642)). This reference was added.**

L90 Should say "… microwindows in the … and … spectral regions."

**Fixed (Line 98-99).**

L76. Was there any quality control and data filtering performed? Please describe. Same for the in situ data.

**No filtering was done for the in-situ (NAPS) data, although all NAPS sites adhere to quality control/quality assurance guidelines set forth by the Canadian Council of Ministers of the Environment (see [https://www.ccme.ca/files/Resources/air/Ambient%20Air%20Monitoring%20and%20QA-QC%20Guidelines_en%20SECURE.pdf](https://www.ccme.ca/files/Resources/air/Ambient%20Air%20Monitoring%20and%20QA-QC%20Guidelines_en%20SECURE.pdf) for details). FTIR columns were retrieved to conform to NDACC standards. Archived species are filtered by RMS/DOFS ratio.**

L110. No need to repeat (National Air Pollution Surveillance Program)

**The repeated bit was a part of the citation for the data (link to an entry references section). This has been removed (Line 119).**

L117. Define the IASI acronym.

**This was previously defined in the introduction (Line 73).**

L121. May not be clear to the reader what a 2 x 2 circular pixel is. Maybe a matrix of 2 x 2 pixels?

**The sentence was replaced with "[a]t nadir, the field of view is a 2 x 2 matrix of pixels, each with a 12 km diameter (Clerbaux et al., 2009)" (Line 132-133).**

L126. Indicate the overpass times of each satellite instrument

**The 3 IASI instruments are onboard the Metop A, B and C satellites which are all in the same polar orbit. Measurements are then performed at 09:30 and 21:30 mean local solar time for the descending and ascending orbits. A sentence clarifying this was added (Line 128-130).**

L155 What do "longer time series" refer to? The length considered in this contribution? Please specify.

**This refers to the duration of the measured dataset. In the method outlined by Weatherhead et al. (1998), measurements that are highly auto-correlated require longer time periods to obtain trends (for any given confidence interval).**

L165. If mirroring a value is the same as taking its absolute value, the readers might be more familiar with the second terminology. It may also be wise to mention that the average of the mirrored residuals include the positive ones.

**The term "mirroring" was used here, as it was also used by Zellweger et al. (2009). The argument for using this terminology is that the residuals should have both negative and positive terms, and in this analysis, the positive ones were "replaced" by the absolute values of the negative ones. The positive residuals are not used, in order to reduce biases introduced by enhancements.**

Fig4. Figure 4 b) seems redundant since no additional information is provided with respect to a).

**This figure was included to better illustrate points made around line 237.**

Fig5. It would seem sufficient to show the correlation plots a) and e) in this figure, while keeping the results of the different resampling periods in the text (L219-223) L300. A larger trend with outliers with respect to that obtained without them may not be conclusive when looking at the data availability of the TOA data series. Measurements seem to be performed more regularly in recent years so to me the increase in seasonal variability is more evident when comparing for example the standard deviations year to year.

**The standard deviations of the TAO columns are in fact increasing (as discussed in Section 3.1). The conclusion was edited to re-iterate this point (Line 338).**

---

## Author Comment (AC2) · 22 Nov 2020

10.5194/amt-2020-319 Revisions: Response to Reviewer 2

Shoma Yamanouchi et al.

General Revisions:

**We thank the two reviewers for their helpful comments, which have enabled us to improve the manuscript. The reviewers' comments are in regular font below and our responses are in bold font. Line numbers in the responses refer to the revised manuscript with changes tracked. Also of note is that there was a minor bug in the trend analysis code; this was revised, and affected values were corrected (this only affected the $2\sigma$ confidence intervals from bootstrap resampling).**

Reviewer 2
**General Comments:**
Why was the nested version of GEOS-Chem over North America not used? It includes Toronto in the domain and is at finer resolution (0.25°ˊ 0.3125°) than the global domain.

**The nested model would be very computationally expensive to run given the long time series. In addition, the amount of storage space required to archive all of the meteorological fields at high resolution for the whole observational record is also a limiting factor. Given that the focus of the paper is the long time series of $NH_3$ observations, we believe that using a model that could be run over the entire time series was more appropriate.**

There is quite a lot of information relevant to model representation of NH3 that is missing in the model description section. These include the following: The inventories used in the model to represent US and Canadian $SO_2$ and $NO_X$ sources that form sulfate and nitrate that influence NH3 uptake to aerosols. The version of EDGAR and whether this is the inventory that represents anthropogenic NH3 emissions over the domain of interest or whether it is a combination of EDGAR and GEIA (now quite outdated and only really used in the model to represent natural NH3 emissions). The base year of each inventory. Whether annual scaling factors are applied to any of the emissions that would have declined due to emission regulations (typically $NO_X$ and SO2). Whether seasonal scaling factors are applied to NH3 emissions in the model.

**EDGAR v4.2 and GEIA were used as global inventories, with GEIA providing the natural source of $NH_3$. The global inventories were replaced with the US EPA National Emission Inventory for 2011 (NEI11) in the United**

States, and by the Criteria Air Contaminants (CAC) from the National Pollutant Release Inventory in Canada. The NEI11 emissions were scaled between the years 2006–2013, whereas the CAC $NH_3$ emissions used 2008 as the base year, with no scaling applied. The NEI11 emissions were hourly, whereas the CAC emissions are monthly.

**The information above was added to the manuscript in Section 2.4.**

The model also seems to be underutilised to provide context for the study region. The inventories could, for example, be used to assess the relative proportion of vehicular, agricultural, and natural emissions to total NH3 emissions and to determine the role of changes in sulfate and nitrate (due to emission regulations of SO2 and $NO_x$ sources) on observed trends in NH3.

**We agree that this would be a good use of the model. However, to effectively attribute the observed change in $NH_3$to vehicular, agricultural, or other emission sources would require use of the nested model, and as we noted in our previous response it is not computationally feasible to run the nested model over the whole observational record. This suggestion would be a valuable follow-up study, focusing on a limited period of the record (e.g., one or two years). Our focus in this manuscript is on the long time series of the FTIR measurements.**

What is the fit that is applied to the data to obtain the trends? And what is the determination of significance? It is stated in the text that "The number of years of measurements needed for the trend to be statistically (2s) significant was found to be 33.8 years and 29.3 years" (p. 6, lines 177-178), but it is not clear why this is the case given that the 2s uncertainty is much less than the trend value. An explicit statement of what the authors use as a significance criterion might help avoid confusion.

**The fit used in this study was a trended Fourier series of order 3. This is discussed in Section 2.6. Two different statistical analysis methods were used in this study. The uncertainties given for each values were obtained using bootstrapping, and the "number of years of measurements needed for the trend to be statistically significant" was estimated using a method outlined by Weatherhead et al., (1998). This method has several drawbacks when used with data with irregular measurement intervals, as is the case for FTIR. This is discussed in the Section 2.6.**

The FTIR instrument and measurements are referred to in figures/tables/text as FTIR, TAO, or TAO FTIR. To avoid confusion, stick with one of these throughout.

Most of the references to the ground-based FTIR were consolidated to simply "FTIR." However, in some sections (especially sections where IASI is mentioned, e.g., Sections 2.3, 2.5, 3.3, 4), the term "TAO FTIR" was used to avoid confusion, as IASI is also an FTIR spectrometer instrument. "TAO FTIR" was also used in places where NDACC, and/or other FTIRs were mentioned. It should also be mentioned that TAO is home to several instruments, including the FTIR. Additionally, in places where the *location* of the FTIR is mentioned (e.g., Figure 7 caption), term TAO was used.

**Specific Comments:**

p1, line 14: There is no context for the use of "resampling" in the abstract to be able to follow what this implies for the results obtained. What is being resampled? And why does it alter the correlation?

**"Resampling" was changed to "averaging" to avoid confusion (Line 14).**

p2, line 38: Briefly elaborate on the link between $NH_3$ concentrations and $SO_2$ and $NO_X$ emissions.

**Sentence clarifying this was added (Line 42-44).**

p2, line 39: "...as well as by reactions with acids in the atmosphere" sounds like it is happening in the gas phase. Make clear that this is a heterogeneous process.

**"[H]eterogeneous" was added to make this clear (Line 43).**

p3, line 59: What is the $NH_3$ source from greenery? Application of fertiliser to gardens and public spaces?

**Chemical fertilizers are "commonly applied" to green spaces in Southern Ontario during spring time (Hu et al., 2018). A statement clarifying this was added (Line 66).**

p4, line 97: What is the shape of the a priori profile used for the retrieval? How does it compare to that from GEOS-Chem?

**The a priori used at TAO is based on the a priori used at Bremen, which is based on balloon-based measurements (Toon et al., 1999). The a priori is comparable to the model profile scaled by a factor of 7. Further details of $NH_3$ retrieval at TAO is described in Lutsch et al. (2016).**

p4, line 121: Odd to express the swath like this. Standard is as 2200 km.

**$2 \times 1100$ was changed to 2200 (Line 131).**

p5, line 137-138: Say what model years are sampled after the one year spin up.
**Fixed (Line 157).**

p5, line 145-147: This approach is reasonable and widespread, but what if the spatial extent is less than the spatial resolution of IASI (at best 12 km at nadir), as seems to be the case in this work?
**As suggested later in the paper, the NH$_3$ column from the FTIR likely has a representative scale of about ~50 km. Also, as discussed in Section 3.3, another FTIR study (Tournadre et al., 2020) found that an FTIR in Paris was capable of providing information about NH$_3$ variability at a ~120 km scales. For these reasons, we believe this methodology is appropriate.**

Figure 2: Does the seasonality differ if the median is calculated for each month?
**There are minor differences, but the general seasonality remains the same; the peak still occurs in May, and minima in January, as was the case when looking at the mean.**

Figure 2: Consider showing the y-axis as 1e16 rather than 1e17.
**This was fixed.**

p7, line 189: Why is the seasonality solely attributed to emissions? What about partitioning of NH3 to acidic aerosols? Is there any seasonality to this process?
**The sentence was revised to "... largely due to agricultural and soil emissions increasing...". A statement about lower NH$_3$ columns during winter, and lower temperatures favoring NH$_4$SO$_3$ was also added (Line 209-211).**

Table 1: Is there a reason that this table is included if this information is already illustrated in Figure 2?
**This was included for completeness, and because while the mean column value of May was given in text, other months were not.**

Table 2: The layout of the table is confusing, as the row labels correspond to specific time periods, but then the final column is labelled "during the same timeframe". What is this timeframe then? Why is the FTIR TAO trend for this same timeframe not given?
**The final column gives the trends of TAO when examining data from the observational periods of NAPS and IASI. The TAO trend for "the same time period" is not given, as it would simply be itself. This was included in the table because this information is given and discussed in text.  The label for this**

**column has been changed to " TAO trends during the same timeframe as either the NAPS or IASI data".**

Figure 4: The lines in (a) are not easy to see. Consider making these thicker.
  **Fixed.**

p12, line 248: Tournadre et al. (2020) is not cited correctly.
  **Fixed (Line 283).**

p12, line 254: What is "simple linear regression"? Ordinary least squares?
  **Yes, this was clarified in text (Line 288).**

p12, line 259-260: It's not clear what this means: "Without temporal resampling, no significant correlation was found (r £ 0.27) for any spatial coincidence criteria". What is this temporal resampling and why does it impact the correlation?
  **As with the p.1 line 14 comment, the word "resampling" was changed to "averaging" to better describe what was done (Line 294).**

Table 3: The information as presented in this table is okay, but would have been more visually interesting and easier to identify patterns in the data if each variable (r, slope etc.) was illustrated on 2D colored grids.
  **This would certainly be visually interesting, but we believe including the numbers is ultimately more important, and we have kept the table as is.**

p13, line 267: What does this gridbox include other than Toronto that might dilute or increase NH3 concentrations and affect the comparison?
  **The gridbox contains areas near Toronto that may increase NH$_3$ due to agricultural emissions, as well as a significant portion of Lake Ontario, which may dilute it.**

Figure 7: It would be helpful to say in the caption or text what this is showing from Table 3.
  **This is mentioned in text (line 287-288).**

Figure 8: It is not easy to discern the red and black points in panel (b).
  **Figure 8b was replotted to make the points easier to discern. Due to the large number of data points, it is difficult to plot them clearly.**

Figure 9: Are units for GEOS-Chem in panel (b) correct?

Yes, they are correct; the GEOS-Chem output was converted to total column values (in molecules/cm$^2$) to allow comparison with IASI measurements.